# A Scalable Finite Difference Method for Deep Reinforcement Learning

## Abstract

Several low-bandwidth distributable black-box optimization algorithms in the family of finite differences such as Evolution Strategies have recently been shown to perform nearly as well as tailored Reinforcement Learning methods in some Reinforcement Learning domains. One shortcoming of these black-box methods is that they must collect information about the structure of the return function at every update, and can often employ only information drawn from a distribution centered around the current parameters. As a result, when these algorithms are distributed across many machines, a significant portion of total runtime may be spent with many machines idle, waiting for a final return and then for an update to be calculated. In this work we introduce a novel method to use older data in finite difference algorithms, which produces a scalable algorithm that avoids significant idle time or wasted computation.

## 1 Introduction

Reinforcement learning (RL) is a sub-field of machine learning that is concerned with finding an optimal policy $\pi^*$ to direct an agent through a Markov decision process (MDP) to maximize an objective $J(\pi)$. In this work, we are interested in methods relating to the *policy gradient* (Sutton et al., 1999) where the policy is parameterized by a set of real parameters $\theta \in \mathbb{R}^d$. These methods search for $\pi^*$ by tuning $\theta$ through gradient ascent on $J(\pi_\theta)$.

Black-box methods for policy optimization are increasingly common in the RL literature (Salimans et al., 2017; Mania et al., 2018; Such et al., 2017), where they can be competitive against more popular approaches under certain conditions (Stulp & Sigaud, 2012a). One such method is *Evolution Strategies* (Salimans et al., 2017) (ES). ES is a distributable learning algorithm that optimizes a population of policies neighboring $\pi_\theta$ by stochastically sampling perturbations from a distribution with mean $\theta$ and maximizing the expected reward of these perturbations. Unlike most purpose-built RL algorithms, ES does not take advantage of the minutiae of the MDP framework, instead leveraging only whole interactions with the decision process to compute updates. In spite of this comparative sparsity of information, ES has been shown to be competitive with powerful learning algorithms like TRPO (Schulman et al., 2015) and A2C (Mnih et al., 2016) in many environments. ES has a particular advantage when transmitting data to asynchronous machines connected to the system is costly because it is able to compress evaluations of a policy in an MDP to a pair of values, one integer and one floating-point, which requires very little network bandwidth to transmit. As a result, the vast majority of the network communication involved in the operation of the algorithm is in the transmission of parameter updates.

One shortcoming of ES is that all information used to compute an update must come from a perturbation of the current parameters. This places severe limitations on the speed at which ES can find an optimal policy because once a sufficient number of perturbations have been dispatched, connected workers must either wait for an update, cut off trajectories early, or have the information they collected discarded. In this work we introduce a method to incorporate information from prior versions of the policy by computing difference quotients even for out-of-distribution samples. This enables workers to compute the return of policies constantly, eliminating idle time.

## 2 Background

In this work we consider the undiscounted-return episodic online reinforcement learning context (see Sutton & Barto (2022, 68)).

### 2.1 Finite Difference Algorithms

In reinforcement learning, finite difference algorithms adjust the parameters of a policy such that an objective $J(\pi_\theta) = \mathbb{E}_{\tau \sim \pi_\theta}[R(\tau)]$ known as the *expected return* is maximized. In this expression, $R(\tau)$ is called the *return* of a trajectory $\tau$, and the policy is a function $\pi_\theta : S \to A$ which maps states of a decision process to actions. The interaction of the policy $\pi_\theta$ and the process produces the distribution of trajectories over which the expected return is defined. To simplify our notation, we abbreviate $J(\pi_\theta)$ to $J(\theta)$.

In this work we are only concerned with finite trajectories, e.g. finite sequences of state, action and reward triples of form
$$\tau = \{(s_0, a_0, r_0), (s_1, a_1, r_1)...(s_T, a_T, r_T)\},$$
created by the interaction between an agent and an MDP starting from $s_0$ and continuing until a terminal state is reached. To maximize $J(\theta)$, the gradient $\nabla J(\theta_u)$ can be used to iteratively tune the parameters $\theta_u$ where $u$ is the number of updates that have been applied to $\theta$ by the learning process. The simplest update rule for $\theta_u$ is

$$\theta_{u+1} = \theta_u + \eta \nabla J(\theta_u),$$

where $\eta$ is a hyper-parameter called the *learning rate*. This update rule is known as stochastic gradient descent, although in the RL setting it is used to maximize an objective rather than minimize one, suggesting "ascent", rather than descent of the function $J$. Finite difference methods estimate the gradient $\nabla J(\theta_u)$ by perturbing $\theta_u$ to form new parameters

$$\alpha = \theta_u + \delta.$$

A trajectory $\tau_\alpha$ is then collected with the resulting policy and reward is computed using

$$R(\tau_\alpha) = \sum_{i=0}^{T} r_i.$$

In this work, only a single trajectory is used to evaluate each perturbed parameter set $\alpha$, so we may refer to $R(\tau_\alpha)$ as $R(\alpha)$ without loss of specificity. The scaled change in reward for a perturbation is then

$$\Delta R = \frac{R(\alpha) - R_{\text{ref}}}{||\delta||},$$

where $R_{\text{ref}} = R(\theta_u)$ in the forward difference case or $R_{\text{ref}} = R(\theta_u - \delta)$ in the central difference (also known as antithetic sampling) case, where a factor of $\frac{1}{2}$ is introduced to account for the alternative method of approximation (Peters & Schaal, 2008; Salimans et al., 2017). A gradient estimate $\mathbf{g}_{\text{FD}}$ can then be accumulated over $N$ perturbations as

$$\mathbf{g}_{\text{FD}} = \frac{1}{N} \sum_{i=1}^{N} \Delta R_i \frac{\delta_i}{||\delta_i||}, \tag{1}$$

where $N$ is a hyper-parameter called the *batch size*.

### 2.2 Evolution Strategies

ES (Salimans et al., 2017) is a black box method for optimizing a distribution of policies that estimates a quantity similar to $\mathbf{g}_{\text{FD}}$ by stochastically perturbing the policy parameters $\theta$ with multi-variate Gaussian noise

$$\alpha = \theta_u + \sigma\epsilon, \tag{2}$$

where $\epsilon \sim \mathcal{N}(0, I) \in \mathbb{R}^{\dim(\theta)}$ and $0 < \sigma < \infty$. The ES gradient estimator is then

$$\mathbf{g}_{\mathrm{ES}} = \frac{1}{\sigma}\mathbb{E}[R(\alpha)\epsilon]$$
$$\approx \frac{1}{\sigma N}\sum_{i=1}^{N} R(\alpha_i)\epsilon_i.$$

However, in practice Salimans et al. (2017) employed an unadjusted form of antithetic sampling in their implementation of ES, which changes $\mathbf{g}_{\mathrm{ES}}$ to

$$\mathbf{g}_{\mathrm{ES}} = \frac{1}{\sigma}\mathbb{E}[(R(\theta_u + \delta) - R(\theta_u - \delta))\epsilon].$$

Notice that, while similar to $\mathbf{g}_{\mathrm{FD}}$, $\mathbf{g}_{\mathrm{ES}}$ does not scale its gradient estimate by the size of each perturbation as in (1), and in the antithetic case it also ignores the usual scaling factor of $\frac{1}{2}$. In spite of these differences, ES still approximates a central-difference estimator of the policy gradient, as shown in recent work (Raisbeck et al., 2020).

ES can be made into a highly scalable distributed algorithm by collecting perturbations $\epsilon$ and their associated rewards $R(\alpha)$ on independent asynchronous CPUs. This enables a learner CPU to collect $(\epsilon, R(\alpha))$ pairs from each worker CPU and compute $\mathbf{g}_{\mathrm{ES}}$ to update $\theta_u$ as soon as a sufficient number of returns have arrived. In addition to the usual advantages of parallel computation, this method of distribution is desirable in cases where network communication is costly because it is possible to compress each pair of data into 2 values, which is significantly less than what workers in other distributed RL algorithms like R2D2 (Kapturowski et al., 2019), SEED RL (Espeholt et al., 2019), IMPALA (Espeholt et al., 2018) and others must transmit to their respective learners.

## 3 Related Work

Numerous studies have investigated the applications of black-box optimization algorithms to RL tasks. Stanley & Miikkulainen (2002) and Stanley et al. (2009) studied approaches to neuro-evolution which evolve both the structure of the policies and their parameters. Hausknecht et al. (2014) successfully applied neuro-evolution to Atari domains. Stulp & Sigaud (2012b) and Hansen et al. (2003) investigated methods of dynamically adapting the covariance matrix of the sampling distribution to facilitate faster learning under various conditions. Sehnke et al. (2010) proposed a method related to ES which estimates a likelihood gradient in parameter space. This work builds from ES (Salimans et al., 2017) which was able to compete with powerful RL algorithms in MuJoCo (Todorov et al., 2012) and Atari (Bellemare et al., 2012) domains. Related to this work is the usage of importance-mixing (Sun et al., 2009) which was applied in conjunction with ES by Pourchot et al. (2018). Their method continually reused information from prior perturbations of the policy so long as they were proximal to the sampling distribution at the current update. Liu et al. (2019) established a method analogous to TRPO (Schulman et al., 2015) which enables sample reuse in ES by optimizing a surrogate objective.

The primary contribution of this work is showing that finite difference algorithms can use information from perturbations which are not proximal to the sampling distribution at the cost of introducing a bias to the gradient approximation.

## 4 Learning Algorithm

A core issue in the implementation of ES is that trajectories in a decision process typically do not require a uniform amount of time to collect; changes in the agent and stochasticity in the environment can lead to dramatic differences in collection time. This means that some workers may take more time than others to return information to the learner, and this asynchronicity could lead to information loss if the learner has computed a new policy by the time a worker finishes testing a perturbation. To address this problem, Salimans et al. (2017) dynamically limited the number of time-steps an agent is allowed to interact with the decision process for before a trajectory is prematurely terminated. While this solution reduces the problem of some machines waiting idle for potentially slow trajectories on other machines to be collected, it introduces a bias to the information used to estimate the reward gradient; the only trajectories with complete information are those that do not get cut off early, which artificially favors shorter trajectories. Further, this approach can only guarantee 50% usage of connected workers in the worst case (Salimans et al., 2017).

### 4.1 Using Delayed Information

To improve worst-case resource use and reduce the bias introduced by early termination, we introduce an approach that enables workers to continually compute and test parameters $\alpha$ without terminating episodes early or discarding data computed using perturbations of previous parameters. To do this, we incorporate returns computed from perturbations of prior policy parameters $\theta_{u-n}$ when estimating $\mathbf{g}_{\mathrm{FD}}$ where $\theta_{u-n}$ is an earlier set of parameters ($0 < n \le u$). This is possible if we treat perturbed parameters $\alpha$ from prior updates $\theta_{u-n}$ as perturbations of the current update $\theta_u$ which have also been biased by the sum $\nu$ of updates to $\theta$ that have been computed over the prior $n$ update steps by the learning algorithm.

We begin with a forward difference estimator of the policy gradient where we perturb the policy parameters in the same manner as ES with $\delta = \sigma\epsilon$,

$$\mathbf{g}_{\mathrm{FD}} = \frac{1}{N} \sum_{i=1}^{N} \left[ R(\alpha_i) - R(\theta_u) \right] \frac{\sigma\epsilon_i}{||\sigma\epsilon_i||^2}. \tag{3}$$

Then, to allow returns from $\theta_{u-n}$ to contribute to $\mathbf{g}_{\mathrm{FD}}$, we treat a reward sampled from a perturbed old policy $R(\theta_{u-n} + \sigma\epsilon)$ as a reward sampled from the current policy whose perturbation $\epsilon$ has been biased.

$$R(\theta_{u-n} + \sigma\epsilon) = R(\theta_u + (\sigma\epsilon - \nu)),$$

where the bias $\nu$ is the difference between $\theta_u$ and $\theta_{u-n}$,

$$\nu = \theta_u - \theta_{u-n},$$

this allows us to treat all perturbations equally

$$\begin{aligned} \alpha &= \theta_u + \sigma\epsilon - \nu \\ &= \theta_u + \theta_{u-n} + \sigma\epsilon - \theta_u \\ &= \theta_{u-n} + \sigma\epsilon. \end{aligned}$$

Note that for $n = 0$ this reduces to the perturbations used by ES in (2). Next we modify (3) to allow for returns from any $\theta_{u-n}$ by replacing the Gaussian noise $\epsilon$ with the biased Gaussian noise $\lambda$ where

$$\begin{aligned} \lambda &= \sigma\epsilon - \nu \\ &= \sigma\epsilon + \theta_{u-n} - \theta_u, \end{aligned}$$

which yields our method to approximate $\nabla J(\theta_u)$

$$\mathbf{g}_{\mathrm{DFD}} = \frac{1}{N} \sum_{i=1}^{N} [R(\alpha_i) - R(\theta_u)] \frac{\lambda_i}{||\lambda_i||^2}. \tag{4}$$

We call this method the *delayed finite difference* (DFD) gradient estimator.

### 4.2 DFD Implementation

We now provide algorithms for the central learner and asynchronous workers for DFD. Data collected from our workers will contain a perturbation, its cumulative reward, the length of the trajectory on which it was evaluated, and the update to $\theta$ that was perturbed, e.g. $R(\alpha)$, $\epsilon$, $T$, and $u$. Note that this is two more values than ES workers must transmit after each episode. To improve consistency in the magnitude of our gradient estimates, we standardize each batch of returns by subtracting the sample mean $\mu_{\mathrm{R}}$ and division by the sample standard deviation $\sigma_{\mathrm{R}}$ of rewards in that batch.

---

**Algorithm 1** DFD Learner

---

    **Input:** $\sigma$, $N$, $T_{\text{lim}}$, update rule
    **Initialize:** $\theta_0$, $T_{\text{total}}$, $u$
    **while** $T_{\text{total}} < T_{\text{lim}}$ **do**
      Transmit $\theta_u$, $u$ to workers
      Compute $R(\theta_u)$
      Collect $N$ evaluations from workers
      Compute batch statistics $\mu_R$, $\sigma_R$
      **for** $i = 0...N$ **do**
        $T_{\text{total}} \leftarrow T_{\text{total}} + T_i$
        $\lambda_i = \sigma \epsilon_i + \theta_{u_i} - \theta_u$
        $R(\alpha_i) \leftarrow \frac{R(\alpha_i) - \mu_R}{\sigma_R}$
      **end for**
      Compute $\mathbf{g}_{\text{DFD}}$ via (4)
      Compute $\theta_{u+1}$ via update rule
      $u \leftarrow u + 1$
    **end while**

---

**Algorithm 2** DFD Worker

---

    **Input:** $\sigma$
    **while** running **do**
      Receive $\theta_u$, $u$ from learner
      Sample $\epsilon \sim \mathcal{N}(0, I)$
      Compute $\alpha = \theta_u + \sigma \epsilon$
      Collect $R(\alpha)$, $T$ from decision process
      Transmit $R(\alpha)$, $\epsilon$, $T$, $u$ to learner
    **end while**

---

In settings where $N$ perturbations can be evaluated by workers faster than the policy's reward $R(\theta_u)$ can be computed, evaluating $R(\theta_u)$ on the learner may result in unnecessary delays at each update while the learner tests the policy. A simple approach to this would be to move the evaluation of $R(\theta_u)$ to the worker such that occasionally $R(\theta_u)$ is collected instead of $R(\alpha)$, but since $R(\theta_u)$ must be known by the learner prior to each update, this would not alleviate the pausing issue. An alternative approach is to approximate $R(\theta_u)$ on the learner as the average of rewards $R(\alpha)$ from perturbations of the current policy instead of measuring $R(\theta_u)$ directly. Note that in cases where this is impossible (e.g. the batch contains only delayed data) or if there is not a sufficient number of perturbations from the current policy to compute a meaningful estimate of $R(\theta_u)$, the average reward over the entire batch of returns can be used as a biased estimate of $R(\theta_u)$ instead. The exact approach we used to estimate $R(\theta_u)$ is described in appendix A.2.

## 5  The Dynamics of Delayed Information

When we consider incorporating delayed information into a finite differences gradient approximation, a pair of questions arise: how does the use of delayed information change our approximation? In particular, how does it affect the quality and bias of the gradient? We answer these questions in several parts: first, we examine the sources of bias in a normal finite differences gradient approximation. Second, we examine the bias introduced by the adjustments made in Salimans et al. (2017) to resolve the efficiency issues mentioned in section 4 and Salimans et al. (2017, 4). Third, we discuss the changes introduced by the inclusion of finite difference partial derivative approximations created using old perturbations. In particular, we discuss the way that this removes a certain kind of bias, as well as the new biases which it introduces, and the increases in the speed of learning suggested by theoretical considerations and empirical results. In the end, we conclude that it is not possible to evaluate the general merits of the adjustment from a purely theoretical perspective. As a result, our theoretical analysis consists of a qualitative description of the effect of various considerations on the performance of an optimizer. For simplicity in this section, we assume that $R$ is a deterministic function of the policy.

While the finite differences algorithm for partial derivative and gradient approximation is well-founded, it is not an unbiased estimator—following from the definition of a differentiable function, finite differences approximations are guaranteed for a differentiable function to converge *in the limit* to the true gradient. For any fixed, perturbation size, the finite differences are only an approximation to the true partial derivative. Interestingly, this is *not* true of the related (Salimans et al., 2017) evolution strategies gradient—that is, although the gradient approximators converge to one another under certain conditions (Raisbeck et al., 2020), the ES gradient approximator is an unbiased estimator of the "search gradient" (Wierstra et al., 2014), while the finite differences approximator is a biased estimator of the true gradient. Importantly, however, differentiable functions are defined by the property that this bias decreases with the size of the perturbations.

Because of the inconsistent rate at which trajectories can be collected from a decision process, performing standard finite differences has an extremely poor worst-case performance, as described in section 4 and Salimans et al. (2017). To resolve this, Salimans et al. (2017) place a dynamically set limit on the length of a decision process. While their method guarantees a worst-case resource utilization of 50%, it also introduces a significant bias: information from episodes which happen to be longer is disproportionately ignored. In RL, where return is often significantly influenced by the length of an episode, has the potential to be a serious problem, above and beyond the normal biases of a finite differences gradient approximator.

That is the state of affairs to which this work responds: is it possible to efficiently perform finite differences without discarding information from some episodes? We have answered in the affirmative, by noting that perturbations of previous sets of parameters $\theta_{u-k}$ can still furnish reasonable approximations of the partial derivative under some circumstances. In particular, by definition there is some radius in which the partial derivative approximations are close enough, for any $\gamma$, and under many circumstances the information from perturbations of earlier sets of parameters will satisfy this requirement. For example, the triangle inequality guarantees that if the distance between the current parameters and a past parameter vector $||\theta_u - \theta_{u-k}||$ and the perturbation size are each smaller than the radius $\beta$ in which partial derivative approximations are $\gamma$-accurate, for a chosen $\gamma$, then these will provide a "good" contribution to the overall gradient estimation. Even when this requirement does not hold formally, well behaved functions (e.g. Lipschitz) often have the property that finite differences with magnitude greater than $\beta$ remain reasonable approximations of the partial derivative within a larger radius about the point.

This contribution, however, comes with a caveat: for a symmetric distribution of returns, such as the Gaussian distributions employed in Evolution Strategies, these perturbations of prior parameters will not be uniformly distributed with respect to direction. Instead, they are directionally similar to the path of updates between $\theta_u$ and $\theta_{u-k}$. In particular, while for any vector $v$, because the distribution of $\epsilon$ is Gaussian, we have for current perturbations $\alpha$

$$\mathbb{E}\left[\langle(\alpha - \theta_u), v\rangle\right] = \mathbb{E}\left[\langle\epsilon, v\rangle\right] = 0.$$

For a delayed perturbation $\alpha'$, we can see that the difference in policy parameters has a non-zero effect:

$$\mathbb{E}\left[\langle(\alpha' - \theta_u), (\theta_u - \theta_{u-k})\rangle\right]$$
$$= \mathbb{E}\left[\langle(\epsilon - \theta_u + \theta_{u-k}), (\theta_u - \theta_{u-k})\rangle\right]$$
$$= \mathbb{E}\left[\langle\epsilon, (\theta_u - \theta_{u-k})\rangle\right] + \mathbb{E}\left[\langle(\theta_{u-k} - \theta_u), (\theta_u - \theta_{u-k})\rangle\right]$$
$$= 0 + \mathbb{E}\left[\langle(\theta_{u-k} - \theta_u), (\theta_u - \theta_{u-k})\rangle\right]$$
$$= -\langle(\theta_u - \theta_{u-k}), (\theta_u - \theta_{u-k})\rangle.$$
$$= -||\theta_u - \theta_{u-k}||^2$$

That is, delayed perturbations have a component which is biased in the direction of the parameters from which they were drawn. In particular, because of the use of an average in the finite differences approximator which we used (and in Evolution Strategies), this means that, even if the delayed perturbations provide good partial derivative approximations, the resulting gradient will be more significant in the direction of the update from which the perturbation was drawn.

Qualitatively, this bias has multiple effects. First, one might imagine that this could serve as a "check", verifying that the parameter changes undertaken improved the performance of the policy. Second, if the updates have worked to improve our function, we might find that incorporation of delayed information acts as a kind of "momentum" (Rumelhart et al., 1986), providing an additional impetus in the direction of previous updates, under the condition that reward has improved as a result of the updates undertaken.

It is difficult to make a general assertion about the net effect of these changes on the performance of the underlying finite differences optimizer from theory alone. In particular, the quality of partial derivative approximations produced by delayed information varies with the return function $R$, the size of perturbations $\sigma$, the parameters $\theta_u$, the number of connected machines, the batch size $N$, and the number of updates which have elapsed since the perturbation was generated, $k$, which is a function of the machines running the worker algorithm. As we demonstrate in the next section, empirical evidence indicates that inclusion of delayed information is often beneficial. This holds *both* when the computational budget (total environment interactions) *and* when the number of updates are held constant—that is, the benefits of including information from delayed perturbations are not *just* in the greater efficiency of the system, but *also* in the direction of the updates themselves, which is likely a combination of two factors: 1) not ignoring information which would be drawn from longer episodes, and 2) the additional "momentum".

# 6 Experiments

To test our method, we studied ES, DFD, and a modification of our method which discards delayed information that we refer to as FD, in 4 of the MuJoCo (Todorov et al., 2012) continuous control environments. Further, we ran PPO (Schulman et al., 2017) under the same conditions and in the same environments as the other methods to provide a point of reference for the performance of our method relative to modern RL algorithms. All methods were tested across the same 10 random seeds. At every update, the policy was used to collect 10 trajectories from the environment. The reward for the policy at that update was then computed as the average cumulative reward over those trajectories. Scores were normalized using min-max normalization relative to the highest average reward over the final 1M time-steps during training and the lowest average initial reward over the first 1M time-steps during training for each environment. All training curves were generated using the RLiable library (Agarwal et al., 2021), which plots the Inter-Quartile Mean (IQM) and point-wise 95% percentile stratified bootstrap confidence intervals over each random seed for each method. Hyper-parameters and full experimental details can be found in appendix A.

## 6.1 Method Performance Study

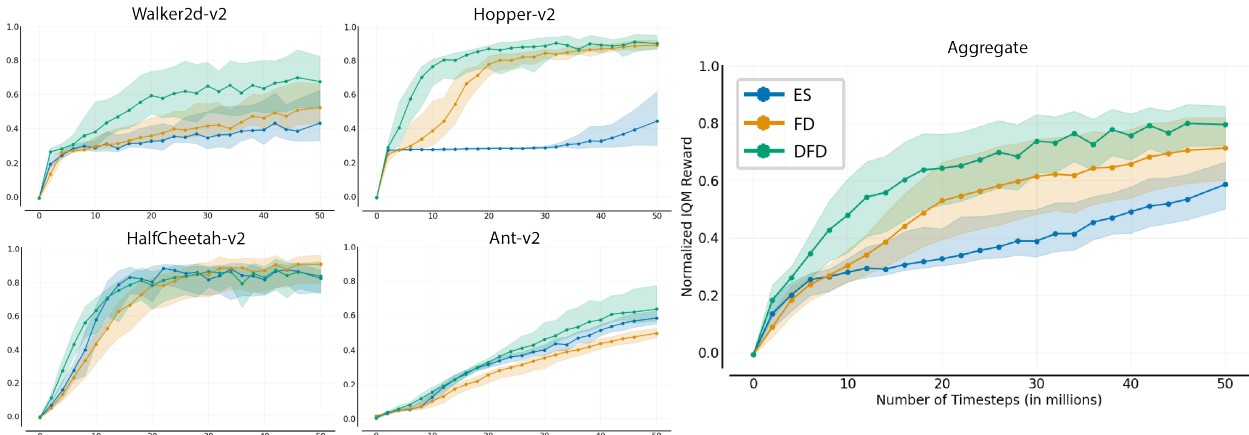

Figure 1: **Methods Comparison** An empirical study comparing ES, FD, and DFD in 4 continuous control tasks. Dotted lines represent the Inter-Quartile Mean and shaded regions show the point-wise 95% percentile stratified bootstrap confidence intervals over 10 random seeds.

We began by testing the performance of ES, FD, and DFD in these environments. We found that incorporating delayed information was typically beneficial to FD, resulting in significantly higher final rewards in two of the four environments. Further, learning appeared to occur faster under DFD than under FD in all environments, as seen in Figure 1.

Table 1: Policy reward over final 1M timesteps per method measured across 10 seeds
mean (standard deviation)

| Environment | PPO | ES | FD | DFD |
|---|---|---|---|---|
| Ant-v2 | 6601.32 (112.02) | 1657.4 (837.4) | 1717.4 (154.4) | 2345.3 (552.4) |
| HalfCheetah-v2 | 9230.85 (991.79) | 4821.9 (697.0) | 5266.2 (759.0) | 5110.0 (811.3) |
| Hopper-v2 | 2062.83 (359.64) | 1812.9 (671.6) | 3313.9 (137.2) | 3392.6 (200.6) |
| Walker2d-v2 | 5815.25 (610.37) | 1720.3 (731.1) | 2108.2 (603.4) | 2495.5 (756.6) |

To provide a point of reference to modern RL methods, we tested PPO under the same conditions as the other methods we examined. The average cumulative reward of the policy over the final 1M time-steps was measured for each method in each experiment, and the results can be found in Table 1. We found that DFD was typically superior to ES and was able to surpass PPO in one environment, though it was worse than PPO in the remaining three environments.

Table 2: Number of updates computed with and without delayed data measured across 10 seeds. mean (standard deviation)

| Environment | FD | DFD |
|---|---|---|
| Ant-v2 | 130.0 (24.9) | 156.3 (29.2) |
| HalfCheetah-v2 | 729.5 (10.1) | 1250.0 (0) |
| Hopper-v2 | 867.1 (10.3) | 1398.1 (22.6) |
| Walker2d-v2 | 941.7 (30.2) | 1446.1 (55.7) |

The benefit of incorporating delayed information when updating the policy may come from the number of updates the optimizer can make within a fixed number of time-steps, the quality of those updates, or a combination thereof. In Table 2 we show the mean and standard deviation of the number of updates computed by FD and DFD in the environments we tested. Including delayed information when computing updates resulted in a 32.8% mean increase in the number of updates computed by the algorithm over the same number of time-steps in these settings. This increase will vary with the length of episodes in the environment, the time it takes for an update to be computed, and the number of workers connected.

## 6.2 Studying the Impact of Delayed Information

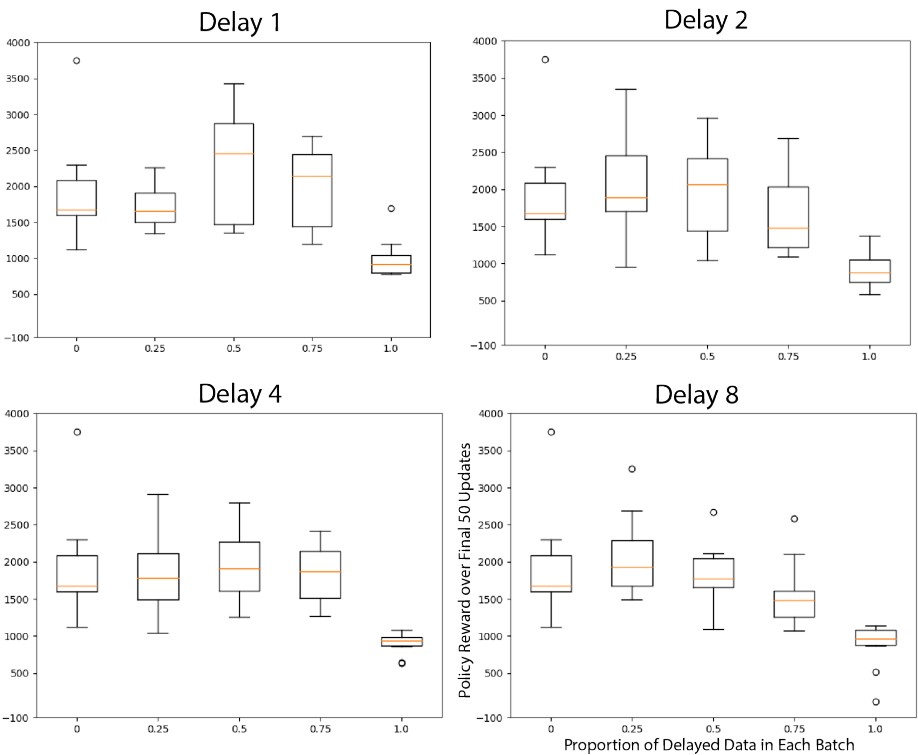

Figure 2: **Empirical Study.** Results of a synthetic experiment in the Walker2d-v2 environment examining the impact that incorporating delayed information has on the learning algorithm. The box-plots in each chart show the distribution of final policy rewards as the proportion of delayed information in each batch increases. At each update a batch of information was artificially constructed containing a fixed proportion of data from perturbations of a delayed policy, where the remainder of each batch was filled with data from perturbations of the current policy. Final policy rewards were measured as the average over the final 50 updates to the policy at the end of training. Each experiment was conducted over 10 random seeds.

To determine how incorporating delayed information impacts the quality of updates in DFD, we conducted a synthetic study where batches of information were artificially held back at each update, enabling us to synthesize batches of data containing fixed proportions of delayed and current information when updating the policy. Our study examined delays of 1, 2, 4, and 8 updates, where at each delay we examined the impact of computing updates with batches composed of either 0% (entirely current) 25%, 50%, 75%, or 100% (entirely old) delayed information. The remainder of each batch was filled with perturbations of the current parameters $\theta_u$ when necessary. This study was conducted over 10 random seeds for 800 updates each in the Walker2d-v2 MuJoCo environment. The final reward of the policy in each experiment was measured as the average cumulative reward over the final 50 updates to the policy.

We found that incorporating some delayed information in each update was not often harmful to the final performance of policies trained during this study. In some cases delayed information resulted in higher quality updates, as shown in Figure 2 when 50% of returns were delayed by 1 update. However, this benefit was sometimes reduced as the proportion of delayed information in each batch approached 1 and the delay increased. In particular, batches containing entirely delayed information resulted in the worst average performance regardless of the delay employed.

# 7 Discussion

Our experiments show that a black-box method can successfully leverage out-of-distribution information when estimating the gradient of an objective. While our method was able to improve over ES and FD, it was still behind PPO in most of the domains we examined. Our method is able to make use of information from distributed workers that are unable to transmit information to a learner before an update is computed, which enables all workers to continually sample and test perturbations of the policy without artificially terminating episodes or pausing while an update is being computed. However, incorporating information that is too old or using a high proportion of delayed information in each batch will reduce the efficacy of the learning algorithm. Practitioners and future researchers should be careful to design systems using DFD such that the rate at which a batch of data can be collected is not significantly faster than the time it takes the learner to compute an update, otherwise it is possible for so much delayed information to be buffered by the learner that it may never catch up to information from perturbations of the current policy. This would result in every batch containing only delayed information, which we found to be the worst performing case in our synthetic tests as shown in Figure 2.

# 8 Conclusion

We have introduced a scalable method for black-box policy optimization using finite differences which is suitable for settings where communication between asynchronous computers is costly. Our method yields notable improvements over ES in continuous control, and does not prematurely terminate trajectories or stop workers from collecting data while the policy is being updated. While we found incorporating delayed data was often beneficial in these settings, their inclusion introduces a bias to the estimation of $\nabla J(\theta_u)$ as discussed in section 5. We found that while the bias introduced to the gradient estimates by delayed information is not always harmful, it can reduce the quality of the learning in some extreme synthetic tests.

There is still a clear gap between refined RL algorithms like PPO and black-box optimizers like ES. However, scalable black-box algorithms like ES and FD still hold some advantages in the distributed compute setting, and enabling black-box methods to make use of out-of-distribution data is a step in the direction of closing the performance gap between the two approaches. Of interest to future work may be combining DFD with the importance-mixing method from Sun et al. (2009), so that perturbations which fall inside the parameter sampling distribution at the current update can be employed multiple times.

Another interesting topic for future work may be investigating different choices for $\delta$. We chose $\delta = \sigma\epsilon$ as in ES where $\epsilon$ is sampled from a multi-variate Gaussian distribution, but different methods for sampling $\epsilon$ may be worth considering. In that vein, one might consider categorically different methods of perturbing the policy, such as constructing perturbations in an agent space (Raisbeck et al., 2021), or the natural space described by Amari & Douglas (1998), rather than the space of parameters.

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

# A    Experimental Details

All experiments were run on c6a.8xlarge Amazon Web Services server instances. For environments other than Ant-v2, we conducted 3 experiments in parallel on a single machine where each experiment was given 4 vCPUs for workers and 4 vCPUs for the learner. For Ant-v2 we tested DFD, ES, and FD in parallel on 3 separate servers using 24 vCPUs each for workers. All experiments with the exception of Ant-v2 were conducted with the fixed hyper-parameters in Table 3. We chose these parameters because they are similar to the equivalent parameters from Mania et al. (2018) and Salimans et al. (2017). Ant-v2 required a significantly larger batch size of $N = 400$ for our method to solve, and was provided 24 workers instead of 4. Each algorithm was tested over the same 10 random seeds.

## A.1    Policy Parameterization

Policies in our experiments parameterized an independent Gaussian for each element in the action vector. Rather than using state-independent variance for each of these distributions, our policies produced both the mean and variance for each action distribution. This means that for an environment with $A$ actions, the policy had $2A$ outputs. The means of each distribution were taken from the first half of a policy's output, and the variances were taken from the second half. The variance of each distribution was linearly transformed from the interval $[-1, 1]$ given by the $\tanh$ activation function onto the interval $[0, 1]$ before the distribution was constructed.

## A.2    Algorithmic Details

Table 3: Hyper-parameters used in our experiments

| Parameter | Value |
|---|---|
| Total Timesteps | 50,000,000 |
| Gradient Optimizer | Adam |
| $\sigma$ | 0.02 |
| $N$ | 40 |
| Adam $\beta_1$ | 0.9 |
| Adam $\beta_2$ | 0.999 |
| Adam $\epsilon$ | 1e-8 |
| Adam $\eta$ | 0.01 |
| Reward Standardization | Yes |
| Policy Architecture | $64 \tanh \rightarrow 64 \tanh \rightarrow \tanh$ |
| Policy Outputs | Diagonal Gaussian |
| Standardized Observations | Yes |
| Observations Clipped | [-5, 5] |
| Random Seeds | 124, 125, 126, ... 133 |
| Worker CPUs | 4 |

PPO was run with 16 parallel workers and the same policy architecture we used in our experiments. Adam's learning rate was decayed from 3e-4 to 0 over the 50M training steps in each environment. All other PPO parameters were set to the values described by Schulman et al. (2017) in their MuJoCo experiments.

Since ES uses antithetic sampling, each noise vector $\epsilon$ constructs two parameter perturbations, $\theta_u + \sigma\epsilon$, and $\theta_u - \sigma\epsilon$. To ensure both methods used the same number of perturbations in each update, we set the batch size in ES to $N/2$ in every experiment.

Following the practice of ES, we maintained running statistics about the observations encountered by any policy during training and used them to standardize each observation by subtracting the running mean and dividing by the running standard deviation before providing an observation to a policy. After standardization, each element of the observation vector was clipped to be on the interval $[-5, 5]$.

As mentioned in the main text, we standardize rewards on the learner at each update using statistics from each batch such that every batch of rewards had zero mean and unit variance. Further, we approximated $R(\theta_u)$ when computing $\mathbf{g}_{\text{DFD}}$ by taking the mean of rewards from perturbations of the current policy in each batch. That is,

$$R(\theta_u) \approx \frac{1}{B} \sum_{i=1}^{B} R(\theta_u + \sigma \epsilon_i), \tag{5}$$

where $B$ is the number of rewards in a batch of size $N$ that were from perturbations of the current policy (e.g. $\theta_u + \sigma\epsilon$). In cases where there are few or no rewards from perturbations of the current policy, we estimated $R(\theta_u)$ as the average reward over the entire batch of data. In our experiments we estimated $R(\theta_u)$ following (5) when $\frac{B}{N} \geq 0.2$, and as the average reward over the entire batch otherwise.

When collecting returns from connected workers the learner continually accepted all available returns until there were at least $N$. If there were more than $N$ returns available, the remaining returns were placed back on the buffer to be collected at the next iteration of the loop.

