# OpenReview forum: "A Scalable Finite Difference Method for Deep Reinforcement Learning"
_TMLR — Rejected by TMLR_

### Review · Reviewer_8KWD · 2022-12-14

**Summary Of Contributions:**

The authors introduce a method to solve the problem of prematurely terminating long trajectories to increase the amount of utilization in evolution strategies (ES) algorithms. Typically, such returns are discarded to maintain the update schedule. However, these long trajectories create delayed returns which can be used to update the policy. The paper investigates how this additional information help in obtaining better policies when not discarded.

The claim of contribution:  Introducing the delayed finite difference method, which allows using delayed returns, achieving higher utilization.


**Audience:**

Yes

**Broader Impact Concerns:**

No concerns.


**Claims And Evidence:**

No

**Requested Changes:**

The following are critical to secure a recommendation for acceptance:
- Use clear and precise language (as discussed in the strengths/weaknesses section).
- Focus on the main contribution of the paper, namely the use of delayed returns. Describe the problem early in the abstract and the introduction.
- Create a fair comparison between algorithms by doing a hyperparameter search. Present rigorous experiments and don’t overstate the results.
- Get rid of the sections/experiments that are less relevant to the paper, as discussed in the strengths/weaknesses section.
- Studying the introduced bias from the delayed returns should be done in this paper, not in future works. Moreover, study the effect of $U_{max}$.
- The limitations of the work need to be discussed.
- The paper needs to add an empirical measure for utilization since it depends on the different variations in the algorithms.
- Correct errors (mentioned in the strengths/weaknesses section) and define the undefined quantities.
- Use clear notations, for example, distinguishing between scalars and vectors.



**Strengths And Weaknesses:**

**Strengths:**
- The idea of incorporating delayed returns into the updates increases the utilization of ES methods, which is beneficial for increasing the number of updates.

**Weaknesses:**
- **The problem is not clearly stated and belatedly introduced.**
  - The problem solved in the paper and the solution the authors provide are not clear from the abstract. The authors mention that they investigate a core problem without mentioning what it is, namely the computational efficiency of such methods. Clear language should be used. Moreover, the problem is not even clearly motivated in the introduction.
  - The problem is understandably introduced in section 3 for the first time in the paper.

- **Unfair comparisons:**
  - All claims and conclusions about how Adam is better than SGD in Fig. 1 may have come from the fact that the authors did not tune the step size. Using a larger step size for SGD would likely improve its performance (this is what likely makes the modified SGD work since it’s essentially SGD with a scaled step size). For a fair comparison, a hyperparameter search must be done to select the best step size for SGD and compare it with Adam.
  - The method of FD-MSGD, DFD-MSGD, and DFD+SGD are not compared in Fig. 3
  - How is the utilization measured? It’s clearly affected by choice of how to compute $R(\theta_u)$. The claim about 100% utilization is not justified. Therefore, empirical analysis of utilization must be added.

- **Irrelevant work added to the main contribution, while leaving out essential work:**
  - The limitations of the work, such as the instability introduced by using returns from old policies, are not discussed. Moreover, the bias introduced by including the delayed returns is not discussed and is left for future work; however, this is a core part of the paper and needs to be discussed and analyzed.
  - The main contribution of this paper is to provide a way of using delayed returns coming from long trajectories. The modified SGDs (MSGD and DSGD) are distracting readers from the contribution. In addition, experiments on Adam and arguments about the magnitude of the update are also unnecessary and erroneous as discussed above. Lastly, the introduction of DSGD is not well-placed in the literature, and there is no mention of any similar methods with a scalar adaptive step size.
  - The authors use $U_{max}$ to prematurely terminate trajectories which goes against the proposed method. A careful study of this parameter is needed.

- **Lack of experimental rigor and overstating the results:**
  - The authors mention that: “incorporating delayed returns is almost uniformly beneficial in the environment we studies.”. However, this statement is not supported by the experiment results shown in Table 2. The delayed finite difference approach only helps in two environments out of four. Moreover, the error bars overlap, so the improvement in these two environments is not statistically significant.
  - It is doubtful that the magnitude of Adam’s updates remains constant. They should depend on the landscape geometry surrounding the current point of parameters which changes with the number of time steps.
  - The suggested method of modifying SGD is unnecessary and introduced based on the wrong analysis of the results from Fig. 1 as discussed in the “unfair comparison” section. There is no need for using the author’s modified SGD when there is a simple solution of using a larger step size.
  - In Fig. 1, the results with Adam, which depends on exact gradients, look identical to FD-Adam, which uses numerical gradients.
  - Fig.1 needs to be plotted on a log scale. The authors mention that SGD updates are not zeros but only appear as zeros since they are small. With a log scale, the results will be clearer.
  - Why are the aggregate results on dissimilar environments useful?

- **Ambiguous and imprecise wordings:**
  - The authors use the words return and reward interchangeably. Each quantity has its own precise definition in reinforcement learning. It is ambiguous to the reader when they are used incautiously.
  - The word bandwidth is used without definition, and it’s unclear from the statements what it means. How does your method, which performs more updates, have low bandwidth? Bandwidth is inversely proportional to the time interval between updates. When we decrease the interval, the bandwidth increases. Moreover, the authors mention that ES methods require very low bandwidth compared with other distributed RL methods. What does this mean?
  - In the text explaining Eq. 5, the authors refer to the quantity $\Delta R$ as the change in reward, where it should be the scaled change in returns.
  - The performance measure IQM is undefined. It’s necessary to show what it stands for and how it is calculated.
  - The authors mention that collecting $R(\theta_u)$ may be costly. It’s not clear why it’s costly compared to $R(\alpha)$.

**Minor issues:**
- The authors used the same notations to write vectors and scalars. Reading these notations would be very challenging to follow for many readers.
- In algorithm 1, it’s missing how the terminal time step for each worker $n_i$ is related to the global time step $u$. Please update the algorithm to fix this.
- It’s missing how the learner makes the workers compute $R(\theta_u)$. The authors mention that $R(\theta_u)$ is computed occasionally without mentioning exactly the frequency of such changes. The authors suggested another modification to their algorithm by approximating  $R(\theta_u)$ by an average over $R(\alpha)$ from all workers. It’s unclear how this way approximates $R(\theta_u)$.
- The step size increment and decrement are denoted by $\epsilon_1$ and $\epsilon_2$, respectively. However, the authors use $\epsilon$ for noise. Please use different notations.
- What is the sum of ranks in Table 2?
- In the introduction, the authors mention that alternative gradient-based rules are often not considered. This statement is clearly wrong since there is an entire research community on finding better optimization methods.
- Consider removing the box surrounding the figures.

---

> ### Author Response · Authors · 2023-01-17
> **Response to Reviewer 1**
>
> Thank you for kind and thorough your review of our work. We have taken your feedback to heart and done our best to address your concerns with our work. In particular, we have removed the portions of our paper dealing with SGD and our modifications of it. We have also removed the $U_\text{max}$ parameter, and included a detailed discussion about the bias imparted to the finite difference gradient estimate when delayed information is incorporated at each update. Further, we have conducted a synthetic study of the impact that incorporating delayed information has on the quality of the policies that our method can produce.

---

### Review · Reviewer_SeGS · 2022-12-21

**Summary Of Contributions:**

The paper contributes to Finite Difference and ES ("à la OpenAI") approaches to RL problems. It brings two contributions:
- it provides a way to use "delayed returns" (information from trajectories obtained with an older version of the policy) into the computation of the Finite Difference Gradient
- from a comparison between Stochastic Gradient Descent (SGD) and Adam, it studies a modified version of SGD which only increases the magnitude of gradient steps and shows that this component if enough to explain most of the superiority of Adam over SGD.
An empirical study with four mujoco benchmarks reveals the (mild) impact of both contributions.

**Audience:**

Yes

**Broader Impact Concerns:**

I see no specific concern about this work

**Claims And Evidence:**

Yes

**Requested Changes:**

* The paper should have a related work section. Doing this will help the authors better delineate their contribution and strengthen it. They should at least have a look at the following papers:

Several papers from the group of K. Choromanski lie in the same domain and consider other families of ESs like CMA-ES and the Cross-Entropy Method (CEM), which do not use a gradient descent component. Reading these papers and a few connected papers will help the authors get a broader view on the relationship between FD methods and ES methods.
- Choromanski, K. M., Pacchiano, A., Parker-Holder, J., Tang, Y., & Sindhwani, V. (2019). From complexity to simplicity: Adaptive es-active subspaces for blackbox optimization. Advances in Neural Information Processing Systems, 32.
- Choromanski, K., Rowland, M., Sindhwani, V., Turner, R., & Weller, A. (2018, July). Structured evolution with compact architectures for scalable policy optimization. In International Conference on Machine Learning (pp. 970-978). PMLR.

Though I cannot claim that the idea of using older trajectories has already been done in gradient-based FD and ES methods, this is closely related to Importance Mixing in the context of CEM or CMA-ES:
- Pourchot, A., Perrin, N., & Sigaud, O. (2018). Importance mixing: Improving sample reuse in evolutionary policy search methods. arXiv preprint arXiv:1808.05832.

In the future work section, the authors mention combining ES and Trust regions methods, they should read:
- Liu, G., Zhao, L., Yang, F., Bian, J., Qin, T., Yu, N., & Liu, T. Y. (2019, July). Trust region evolution strategies. In Proceedings of the AAAI Conference on Artificial Intelligence (Vol. 33, No. 01, pp. 4352-4359).

These papers are starting points, the authors must also look for closely related papers.

* The authors should better consider whether their two contributions are related and they should be more explicit in the introduction about the articulation between these contributions

* About the DSGD contribution, it could be presented either as a dissection of the sources of efficiency in the Adam mechanisms (as currently put forward in the discussion) or as an alternative to Adam, in which case the paper should state more clearly if there are unquestionable advantages in using DSGD instead of Adam (in terms of bandwith, wall clock time efficieny or whatever...) and this advantage should appear in a study.

* We can see immediately from the first paragraph of the introduction (sentence "Most model-free ...") that the authors do not have a strong expertise in non-ES-based RL methods. What are "algorithms like Q-learning"? Do they mean "derived from dynamic programming " (as mentionned in the second paragraph)? Also, the choice of references in the sentence is rather weak and there would be more to say. My guess is that the authors added this perspective because they felt that speaking about "true" RL is necessary to be in the scope of TMLR, but I do not share this feeling: we see more and more "pure ES" papers at NeurIPS, for instance. So I encourage the authors to speak only about what they are expert of. In the future work I had the same feeling, the authors may mention the relationship between reusing old trajectories and using a replay buffer, but I don't believe they are ready to cover such topics in depth.

* Unless I'm wrong, Figure 3 contains all the information already present in Figure 2. The authors could choose three blue-to-green colors for the curves of Figure 2 and three red-orange-pink colors for the additional ones and put everything into a single figure.

 * I understand the argument of the authors about comparing the best performing policies, but it has been mentioned several times in the field that this is a bad practice as this performance indicator is very fragile (see e.g. the Henderson paper, Deep RL that matters): run it a second time with different seeds and you will most probably get different results...

* The greater sensitivity of Adam to initial conditions could be put more forward

* In the abstract "investigate a core problem": you should rather say which problem you investigate and let the reader decide if it is core or not.

* In 3.1, remove all "will". You already did it, using the present is better in science.

* For equations, use \eqref{label}, which produces (number).

* p4, the paragraph starting with "Collecting R(\theta_u)..." is still a little unclear to me. Could you be more explicit about the collection process? With a schema?

* Algorithms 1 and 2 are not referenced in the text, all floats should be referenced. Eventually, the algorithms could be explained.

* In Figure 1, shouldn't you use a log scale to reveal that SGD is not at 0? What is IQM in the y-axis label (also present in figs 2 and 3)?

* How did you compute the magnitude of gradient updates?

* The "sum of ranks" column in Table 2 is neither commented nor explained

* I would remove the discussion and use it rather as an articulation of the second contribution (see above)

* the future work section should be moved in the end of the conclusion (without a title)


**Strengths And Weaknesses:**

Strengths:
- both contributions make sense and are presented rather clearly
- the empirical study uses rliable, hence avoiding some of the common pitfalls in many RL comparisons

Weaknesses:
- most importantly, the paper lacks a related work section, as a result it ignores a lot of relevant work
- some readers might find some of these contributions too straightforward to be of interest. Honestly, I'm not sure that these ideas have never been published before
- the two contributions are rather independent, they could have been presented in two different papers and it is unclear if their combination brings something special with respect to using them separately
- the "story telling" of the paper could be much improved.

All in all, my general feeling is that this paper may be turned into a useful contribution provided a significant amount of work, but under its current form it lacks a lot of maturity, it is far from being ready to be published

---

> ### Author Response · Authors · 2023-01-11
> **Response to Reviewer 2**
>
> We thank you for your kind and thorough review of our work. We have taken the feedback from all our reviewers and are working to make the following changes to the manuscript:
>
> 1. We will excise our contributions relating to SGD such that the paper is entirely about DFD.
> 2. Our discussion of the bias imparted to the finite differences gradient estimation in Appendix C will be expanded and incorporated into Section 4.
> 3. We will conduct an experiment which empirically demonstrates the impact that incorporating delayed returns has on the training process for various delays and proportions of delayed returns in each batch.
> 4. We will discus a larger set and greater variety of related works.
> 5. The abstract and introduction will be adjusted to more clearly outline the problem, motivation, and proposed solution.
> 6. Table 2 will either be more clearly explained or removed.
> 7. The $U_{\text{max}}$ parameter will be removed.
> 8. The future work sub-section will be moved to the end of the conclusion and the discussion section will be modified to describe the shortcomings of the proposed method as well as its benefits.
> 9. The narrative structure of the paper will be made more cohesive.
> 10. Rather than using the word "return" when referring to the data used by FD methods to compute updates, we will choose a word which does not conflict with existing Reinforcement Learning terminology.
> 11. Algorithms 1 and 2 will be explained in further detail.
> 12. We will use PPO under the same conditions we tested DFD to provide a baseline for readers who are already familiar with traditional DRL methods.
>
> We have heard your feedback about related works and will adjust the manuscript accordingly. Regarding the specific works you mentioned, the importance mixing method stands out as being directly related to our delayed returns contribution. Are there any other parallels in those works you would like to see us address directly?

---

> ### Comment · Reviewer_SeGS · 2023-01-12
> **Error in responding?**
>
> Is it on purpose that the authors copy-pasted the same answer to two different reviews?
>
> Also, the answer is specifying what the authors may do some  day in the future, but they should rather submit a revision, and describe in their answer what they have done, right? (the action editor may correct me here if I misundertood the TMLR review process).

---

> > ### Author Response · Authors · 2023-01-12
> > **Clarification**
> >
> > Hello Reviewer 2,
> >
> > We hope that you will acept our apologies for the inclarity in our response yesterday - we are in the process of completing the listed revisions but wanted to deliver a response and question to you as soon as possible.
> >
> > We would be grateful if you could spare the time before the deadline for our revised version of the paper to review the list of forthcoming changes and let us know if we have missed any of the major concerns of your review (we have not listed many minor details of the revision), and answer the question at the end of our comment.
> >
> > Thank you again for the time you have spent in this process and for your kind and thorough review,
> > Authors

---

### Review · Reviewer_tydB · 2023-01-03

**Summary Of Contributions:**

The authors of this paper investigate and address a core problem of utilizing black-box optimization algorithms (e.g., evolutionary strategy) for deep reinforcement learning.

Specifically, the authors leverage finite difference methods to derive a delayed finite difference (DFD) gradient estimator, which can address the sample-inefficency issue of previous work using ES (Salimans et. al, 2017).

Further, the authors present the performance of incorporating the estimated gradient into the Adam optimizer and several modified formulations of stochastic gradient descent (modified SGD and dynamic SGD). The empirical performance shows promising results and significant improvements over naive black-box optimization methods (e.g., ES), which may further motivate researchers to study the benefits of the proposed techniques in this paper.

**Audience:**

Yes

**Broader Impact Concerns:**

N/A.

**Claims And Evidence:**

Yes

**Requested Changes:**


- Add ablation study on the critical hyperparameters, such as $\delta$.

- If possible, please add an error analysis figure to show the approximated error using finite difference methods.

- If possible, please add the training time and sample utilization of your methods and ES, which would further demonstrate the advantage of your method.

- If possible, please discuss why you would like to introduce modifying SGD and dynamic SGD. And also I think it is also to incorporate the strategies into Adam. and how would that actual performance be?

- Please add PPO to your experiments to help people understand the current stage of black-box-based optimization methods.

**Strengths And Weaknesses:**

Strengths:

- The paper presents a scalable method to address the sample inefficiency problem of previous DRL using black-box optimization methods, where ES can not leverage all the samples collected to improve the learning policy. I think this is a critical problem, and I have not seen many previous related works trying to address the issue.
- Since finite difference methods are approximated approaches to estimating the gradient, it is also good to utilize adaptive step size methods such as Adam to improve performance further. And the experimental results demonstrate the benefits of utilizing such adaptive step size strategies.
- The authors present the experiments using recent proposed systematic evaluation methods (Agarwal et al., 2021), which is important since most of existing black-box based estimation methods tends to have high variance.
- The several directions discussed in the conclusion are promising and interesting, especially given the promising results presented in this paper.

Weakness:

- As far as I know, even though this paper improves the performance of black-box gradient-based methods, its performance is still far from gradient-based methods such as PPO or off-policy-based methods such as SAC.

- The motivation of proposed modifying SGD and dynamic SGD should be discussed more. I can not see the advantage of using these two methods (if I missed anything, please correct me :) ).

- Some ablation studies should also be conducted to help readers understand the proposed method's limitations and robustness. For example, how can we choose $\delta$ in these environments? Some important hyperparameters should be studied and discussed to help readers to understand.

- Also, since the finite difference is an approximated method, it would be great to see the approximated error between the approach and the true gradient estimation.

---

> ### Author Response · Authors · 2023-01-11
> **Response to Reviewer 3**
>
> We thank you for your kind and thorough review of our work. We have taken the feedback from all our reviewers and are working to make the following changes to the manuscript:
>
> 1. We will excise our contributions relating to SGD such that the paper is entirely about DFD.
> 2. Our discussion of the bias imparted to the finite differences gradient estimation in Appendix C will be expanded and incorporated into Section 4.
> 3. We will conduct an experiment which empirically demonstrates the impact that incorporating delayed returns has on the training process for various delays and proportions of delayed returns in each batch.
> 4. We will discus a larger set and greater variety of related works.
> 5. The abstract and introduction will be adjusted to more clearly outline the problem, motivation, and proposed solution.
> 6. Table 2 will either be more clearly explained or removed.
> 7. The $U_{\text{max}}$ parameter will be removed.
> 8. The future work sub-section will be moved to the end of the conclusion and the discussion section will be modified to describe the shortcomings of the proposed method as well as its benefits.
> 9. The narrative structure of the paper will be made more cohesive.
> 10. Rather than using the word "return" when referring to the data used by FD methods to compute updates, we will choose a word which does not conflict with existing Reinforcement Learning terminology.
> 11. Algorithms 1 and 2 will be explained in further detail.
> 12. We will use PPO under the same conditions we tested DFD to provide a baseline for readers who are already familiar with traditional DRL methods.
>
> Regarding your request that we study different choices of $\delta$ in our method, we don't think we can meaningfully conduct such a study with sufficient rigor before the final decision deadline at the end of this month. While we think studying different choices of $\delta$ is of interest to FD methods (as alluded to in section 6.1 of the current manuscript) the complexity of choosing an appropriate set of sampling methods and testing each of them sufficiently is not feasible in the remaining time.

---

### Comment · Reviewer_SeGS · 2023-01-20
**The authors must improve their journal review answering behavior**

The authors do not seem to know a few basics of the adequate behavior when answering to reviews in a journal:

- the reviewers have spent time elaborating their review. They appreciate if the authors can answer each of their points individually, saying whether they agree, if not explaining why and if yes being precise (where and what) about the changes they make in the manuscript to take the reviewer's point into account.

- it is a good practice when submitting the new version of the manuscript to highlight all the changes that have been applied, for instance by using colors. This helps the reviewers saving a lot of time, and time is precious.

Because the authors failed on the above two points, I have to read the whole paper again as a first submission. This may affect my evaluation with a negative bias. I will do my best to ignore this negative bias when submitting my recommendation, but I want the authors to know so that they can do better next time.

---

> ### Author Response · Authors · 2023-01-24
> **Thank you for your patience**
>
> Thank you for your patience and explanation of our shortcomings. This is our first submission to OpenReview, and we are still learning how best to engage with its review discussion process. In the future we will improve on this by outlining the relevant sections of text that are changed in each revision and addressing each point of feedback individually. We are grateful for your feedback, and sincerely apologize for the inconvenience we have caused.
>
> Regarding your comment titled "Need for further improvements", the materials in the email we received from OpenReview after the final response to our first submission was posted seemed to indicate that the remaining time until the final decision date is meant for reviewers to examine the revision we have provided rather than for us to make further changes to the text, so we do not believe we are allowed to submit another revision addressing your most recent comments before the final decision deadline. Regardless, we appreciate your feedback about our most recent revision, and we would be happy to provide a detailed response both to that review and your prior review of our initial submission if that would be appropriate.

---

### Comment · Reviewer_SeGS · 2023-01-20
**Need for further improvements**

After reading the new version of the paper, I found the paper has significantly improved, but I have a few comments that may help the authors further improve their work.

- to me the main weakness is in Section 5 which I found rather unclear. I think this section should be either shortened a lot or strengthened with more elements from the literature. The only message that I retained from reading the long text was that a theoretical study of these matters was impractical so we can only rely on an empirical study. In that respect, Fig. 2 is a good contribution of the paper.

- another weakness is the related work section which is still quite out of focus with respect to the current work. The authors should look for papers trying to remove the idle time in worker-learner architectures with alternative methods such as:

Lee, K., Lee, B. U., Shin, U., & Kweon, I. S. (2020). An efficient asynchronous method for integrating evolutionary and gradient-based policy search. Advances in Neural Information Processing Systems, 33, 10124-10135.

or try to find papers studying the impact of old samples in policy gradient methods (there must be many of them about Importance sampling, start searching from the TRPO paper).
Generally speaking, for the mentioned works the authors should be able to tell in what respects their method differs from the one they mention. And maybe the related work could come after the methods.

Alternatives: instead of collecting a single trajectory and wait, the workers could collect as many trajectories as then can until thet are asked to stop. So different workers would compute their score based on a different number of trajectories. Is there a bias issue with this approach?

More local points:

- maybe you could add a worker-learner architecture figure earky in the paper to make the point clearer?

- p4, last line: division -> dividing?

- p6, the sentence: "In RL, where return is often significantly influenced by the length of an episode, has the potential to be a serious problem, above and beyond the normal biases of a finite differences gradient approximator."
makes no sense. Something is missing. This has contributed to my poor undersanding of Section 5.

- p6 bottom: "when ... and when ..." -> (remove second when, there is no verb in the first part)

- I would move the content of page 13 (details of the methods) in the main text, and have the four subplots of Figure 2 aligned in one row if saving space is needed

- Table 1: put the highest scores over a row in bold

- Figure 2 cannot be completely understood just from the caption (it should). Put a label on the x-axis (such as delayed information rate) and y-axis (performance). Most importantly, explain Delay 1, 2, 4, 8 in the caption.

---

### Decision · Action_Editors · 2023-03-02

**Recommendation:** Reject

**Comment:**

See comments on Claims and Evidence above.

**Audience:**

I think all the reviewer's agree that "some individuals in TMLR's audience" would be interested in the findings of this work, if it's claims were convincingly supported.

**Claims And Evidence:**

The reviewers all agree that the updated manuscript has been a significant improvement in regards to the paper's claims and evidence for those claims.  There are still, however, shortcomings in the article's evidence for its claims.  Here's some notable comments (not found in the additional reviewer feedback).

* "I still don't think the authors have a detailed discussion of their approach and previous work. There is no comprehensive discussion of why their method is better while others do not, and what is the main different between their work and previous ones."

* " the comparisons were not fair since the step size was not tuned for different methods. It is unclear what conclusion can be drawn from the current experiments. Having rigorous experiments is critical in securing recommendations for acceptance."

Of the remaining concerns, while I consider the shortcomings in comparisons to other work to still be relevant to the paper's claims and evidence, it is also an area the largely can always be improved.  The more fatal shortcoming is the second bullet point above.  Careful attention should be made to reviewer 8KWD's concerns about whether the empirical results do provide evidence for the paper's claims.  In particular, giving focus to fair hyperparameter tuning.

As a decision needs to be made on the paper in its current form, these remaining shortcomings suggest the paper's claims are not "supported by accurate, convincing, and clear evidence".